# Propaedeutic and Therapeutic Practices Used for Retained Fetal Membranes by Rural European Veterinary Practitioners

**DOI:** 10.3390/ani14071042

**Published:** 2024-03-29

**Authors:** Christian Hanzen, Hamza Rahab

**Affiliations:** 1Fundamental and Applied Research for Animal and Health Department, Veterinary Faculty, University of Liège, B43, B 4000 Liège, Belgium; 2Animal Health Team, Biotechnologies and Health Division, Biotechnology Research Center (CRBt), Constantine 25000, Algeria; h.rahab@gmail.com

**Keywords:** retained fetal membranes, survey, cattle, practitioner, diagnostic, treatment

## Abstract

**Simple Summary:**

A cow is considered to have a retained placenta (RP) when the fetal membranes are visible at the vulva or identified in the uterus more than 24 h after calving. The problem of retained fetal membranes concerns dairy cattle more than beef cattle. This pathology is a major risk factor for uterine infections, ketosis and mastitis. It delays the postpartum resumption of cyclic ovarian function. This pathology also has economic consequences relating to increased veterinarian costs, the risk of culling rate, decreased milk production and reproduction performances. The main observations resulting from our survey of 700 practitioners across five European countries are as follows: abortion and twinning remain the main causes of retained fetal membranes; vaginal exploration remains the main diagnostic method used by veterinarians; more than half of veterinarians attempt to manually remove the placenta; tetracyclins, cefapirin and penicillins are the most used intrauterine antibiotics; and PGF2α, NSAID, homeopathy and oxytocin are used by a huge number of veterinarians. Our study confirms the necessity to improve and rationalize the diagnostic and therapeutic approach of the RFM, mainly in the context of the different measures to reduce the important problem of antibiotic resistance.

**Abstract:**

The present study aimed to monitor the practices of European veterinarians for the diagnosis and treatment of retained fetal membranes in cattle. A questionnaire was established and distributed to veterinarians from five European countries. A total of 700 veterinarians participated in the survey. A vaginal examination, general examination and uterine palpation are carried out by 71%, 38% and 23% of veterinarians, respectively. Moreover, half of the veterinarians attempt to remove the placenta manually, 70% of them administer a combined local and general treatment if the cow has a fever (more than 39.5 °C), and 50% of them only administer IU treatment if no fever is observed. Tetracyclins, cefapirin and penicillins are the most used intrauterine (IU) antibiotics, whereas penicillin is the most used parenteral one. All other European veterinarians were less likely to use cefapirin and more likely to use oxytocin, Ca perfusion and NSAID than French and Walloon veterinarians. In conclusion, our study confirms the necessity of improving and rationalizing the diagnostic and therapeutic approach of the RFM, mainly to reduce the important problem of antibiotic resistance.

## 1. Introduction

Retained fetal membranes (RFMs) are one of the most common pathologies occurring after parturition, especially in dairy cattle. It is a major risk factor for puerperal metritis (PM) and endometritis, ketosis and mastitis [1,2,3,4,5]. It delays the postpartum resumption of cyclic ovarian function and increases the interval from calving to first ovulation [6]. This pathology also has economic consequences relating to increased veterinarian costs, the risk of culling, decreased milk production and reproduction performances [7,8,9,10]. The economic cost has been estimated to be USD 297 per treated case [11].

A cow is considered to have a retained placenta (RP) when the fetal membranes are visible at the vulva or identified in the uterus more than 24 h after the first observation of the cow or heifer following calving [7,8]. As reviewed by Kelton et al. in 1998 [12], the reported frequency of RFM, based on 50 citations from 1979 to 1995, ranged from 1.3% to 39.2% [13,14]. The median lactational incidence rate was 8.6% [12].

Numerous risk factors for RFMs have been described and reviewed [7,10,15]. Among these factors, dystocia, the birth of a male or a dead calf, decreased pregnancy length (abortion or calving induction), twinning, hypocalcemia, an increased number of pregnancies, summer and reduced vitamin E supplementation during the dry period, are notable [16,17,18]. Directly or indirectly, all of these factors are associated with failure in the timely breakdown of cotyledon–caruncle attachment after delivering the calf [7,19].

The physiological events regulating the separation and expulsion of fetal membranes (maturation process) are relatively complex and involve molecular and hormonal events, immune and inflammatory responses and extracellular matrix remodeling. Steroid hormones and prostaglandins predominantly govern all these factors and processes. A better understanding of them could offer the possibility of developing strategies for the treatment and prevention of RFM [19].

Different therapies have been proposed to deal with RFM in cows [7,20,21,22]. However, the efficacy of many of these treatments is questionable. Classically, the placenta is manually removed in association with intrauterine treatment [23,24] or by injecting antibiotics if the cow is febrile [25,26]. Some studies have shown that intrauterine manipulation decreases uterine defense mechanisms [21,27] and impairs subsequent fertility [28]. The use of antibiotics administered as uterine infusions or boluses in the treatment of RFM has demonstrated conflicting results [21]. Tetracycline, inhibiting the matrix mettaloproteinases, might interfere with normal placental detachment mechanisms [29]. Ceftiofur has been evaluated in many clinical trials on RFM and puerperal metritis treatments [25,26,30]. Unfortunately, this third-generation cephalosporin can no longer be justified, and its use should be restricted unless an antibiogram indicates its need [31]. Furthermore, the administration of antibiotics in livestock should be minimized to reduce the prevalence of resistant bacteria [32]. The overdosing of antibiotics, which is frequently practiced in the field, increases the risk of residues being present in milk [33].

It seems that prostaglandins and oxytocin have no effect in terms of preventing or treating RFM [34,35,36]. Due to its germicidal and immunity-stimulating effects [37,38,39], ozone could be recommended as a non-antibiotic intrauterine treatment for RFM [40].

Due to the diversity of approaches proposed for use in the diagnosis and treatment of RFM and the absence of conventional treatment, different treatment approaches may exist in the field. However, little is known about the state of RFM and its management under field conditions in Europe. Thus, the aim of our study was to describe the approaches adopted by rural practitioners from different European countries to diagnose and treat cows with RFM.

## 2. Materials and Methods

### 2.1. The Questionnaire

The 14 questions of the questionnaire (Appendix A) consisted of multiple choice and 3-point Likert scale questions, i.e., always, sometimes (meaning times to times according to the clinical situation of the animal) or never and was developed on the online platform Survey Monkey. It was divided into 4 sections: (i) the first section (5 questions) concerned social demographic variables, (ii) the second (3 questions) concerned the definition, estimated prevalence and causes, (iii) the third (1 question) concerned the methods of diagnosis and (iv) the fourth (5 questions) concerned the therapeutic strategies used.

The questionnaire was shared through the e-mail listings of the veterinarian associations of France, Belgium (Wallonia and Flanders), Sweden, Austria and The Netherlands.

### 2.2. Data Analysis

Data from the returned questionnaires were downloaded in Excel format and analyzed using IBM SPSS Statistics, Version 23.0 (IBM Corporation, Cary, NC, USA).

Descriptive analysis was carried out to describe the dataset and outline the management practices utilized by veterinarians to define and diagnose RFM, as well as their attitudes toward its control.

Ordinal regression was carried out to assess the impact of veterinarian characteristics (explanatory variables) and diagnosis and treatment methods (outcome variables). We first used univariate analysis via Khi2 to select variables that were possibly associated between the explanatory and outcome variables. Variables with *p* < 0.2 in univariate analysis were subjected to multivariate ordinal regression. The model was built using a backward Wald selection method, and explanatory variables with a value of *p* < 0.05 were considered significantly associated with the outcome and retained in the final model. The predictive capacity of the model was assessed using the Nagelkerke Pseudo-R-squared.

## 3. Results

The total number of completed answers was 682.

### 3.1. Social Demographic Variables

The majority of the respondents were male (76.5%) and worked in France (47.5%), Belgium (20.6%), Austria (20.4%), Sweden (6.5%) or the Netherlands (5.0%). Their years of experience were as follows: <10 years (26.1%), 10 to 20 years (25.5%) and >20 years (48.4%). The majority of veterinarians (61.2%) have more than 50% of dairy herds in their practice and are (49.4%) mostly bovine-oriented (>80% of their time devoted to bovines).

### 3.2. Characteristics of RFM

The large majority of practitioners define a case of RFM as the non-expulsion of the placenta beyond 24 h (48,4%) compared to other categories (<6 h: 5.4%, 6–12 h: 38.9%, 12–24 h: 48.4% and >24 h: 7.3%). The majority of veterinarians (76.1%) believe that the prevalence of RFM is similar in beef and dairy cattle. Practitioners indicated that, in dairy cattle, the frequencies of RFM are lower than 5%, between 5 and 10% and higher than 10% in 22.0%, 46.6% and 21.4% of cows, respectively. In beef cattle, such frequencies are 49.9%, 26.8% and 6.1% of cows, respectively. In dairy and beef cattle, 10 and 17.2%, respectively, have no idea of such prevalence (Table 1).

When analyzing the factors responsible for RFM in cows, 59 and 59.8% of veterinarians estimated that abortion and twinning are the two main factors involved. Excessive or insufficient BCS is more often proposed for dairy cattle than beef cattle (30.0% vs. 18.0% and 15.3% vs. 9.7%). An excess of minerals before calving was more often evoked in dairy cattle than in beef cattle (36.2 vs. 25.3%). In beef cattle, it seems that Cesarean section (C-section) is more often responsible for RFM than in dairy cattle (14.4 vs. 9.1%) (Table 1).

### 3.3. Diagnosis of RFM

Vaginal examination (VE) is the method most systematically used by the veterinarians (71.0%). Such an examination is usually carried out after cleaning the vulva with an antiseptic solution (AS) (30.6%) with (25.1%) or without (21.1%) water. In this study, 38% of the veterinarians indicated that they systematically perform a general examination (GE) of the animal, and 23.7% indicated that they perform a uterine manual examination through the vagina (MP). Very few veterinarians (1.2 to 3.9%) declared systematically evaluating non-esterified fatty acids (NEFAs) or betahydroxybutyrate (BHB) in the blood, urine or milk. The body temperature of the animal is rarely systematically taken by the farmer (3.1%) (Table 2).

Very few veterinarians (17.3%) recommend that the farmer take the temperature of the cow during the days following RFM.

### 3.4. Therapeutic Approach

An attempt to remove the placenta manually is always performed by 48.7%, sometimes by 41.9% and never by 9.4% of the veterinarians, respectively.

Most veterinarians treat cows with RFM even if the animals do not present with fever (91.1%), and local treatment is more often given (52.1% of respondents) in this case. In the case of fever, a combined local and systemic (70.4%) treatment is more often used than locally (15.6%) or systemically (28.6%) alone.

Considering treatments using intrauterine antibiotics, tetracyclins, cefapirin and penicillins are more often systematically used than other types of antibiotics. Moreover, antiseptic solutions (24,3%) or povidone–iodine or iodine solutions (13.7%) are sometimes used. For the systemic route, penicillins are most often used (22.1%). Third- and fourth-generation cephalosporins are sometimes used by 43.4% of the veterinarians (Table 3).

For non-antibiotic treatments, PGF2a and nonsteroidal anti-inflammatory drugs are used systematically by 14.5% and 12.3% of veterinarians, respectively, and occasionally by 48.0% and 46.1% of veterinarians, respectively. Occasional use is also true for homeopathy (29.7%), oxytocin (34.0%) and calcium perfusion. Other drugs are always used by less than 5% of the veterinarians.

A control visit during the next three weeks after RFM is always or sometimes carried out by 38.1% and 49.2% of the veterinarians, respectively.

### 3.5. Ordinal Regression Analysis

The results of the ordinal regression analysis are presented in Table 4.

All other European veterinarians were more likely than French and Walloon veterinarians to recommend that a farmer take the temperature of a cow when dealing with the case of a retained placenta case.

All other European veterinarians were less likely than French and Flemish veterinarians to use no treatment when the cow has no fever. Moreover, veterinarians with more than 20 years of experience were more likely than those with less experience to do so.

All other European veterinarians were less likely to use cefapirin and more likely to use oxytocin, Ca perfusion and NSAID than French and Walloon veterinarians. Moreover, veterinarians with more than 20 years of experience were less likely than those with less experience to use NSAIDs.

## 4. Discussion

There is no consensus on the definition of RFM: 38.9% of the practitioners consider a delay of 12 h, and 48.5% consider a delay of 24 h. In a recent survey, 71.8% of Belgian practitioners consider a delay of 24 h [41]. Such differences can be the consequence of different universities of graduation. A delay of 24 h is usually considered by the majority of authors [7,8].

The estimated prevalence of RFM considered by the respondents is higher in dairy than in beef herds. This is in accordance with field studies, where the median lactational incidence rate was 8.6% in dairy cattle [12] and 3.5% in beef cattle [42]. Factors relating to metabolic stress (BCS < 3 or >4, excess of minerals) might explain this difference. Moreover, during the transition period, many high-producing dairy cows experience reduced feed intake, negative energy balance and hypocalcemia. All of those conditions modify the immunological mechanisms of placental maturation through a reduction in neutrophil function, which is impaired by a decrease in vitamin E and Se supplementation or by hypocalcemia indirectly caused by high calcium diets prior to calving [16,43,44]. The majority of practitioners identified abortion and twinning as the two main causes of RFM. In one study on 160,188 Maas Rhin Yssel cows, Joosten observed that RFM is observed in 61.6% of abortions (<260 days) and in 36.8% of multiple births [18], respectively.

Not surprisingly, the majority (71%) of practitioners perform a vaginal examination in the case of RFM. Such an examination is necessary to confirm the diagnosis when the fetal membranes are not externalized [45] or in 48.7% of cases to attempt to manually remove the placenta.

The manual removal of the placenta remains common practice [21,46] despite several studies that fail to demonstrate a beneficial effect on reproductive performance or milk yield [25,28,47]. This practice is quite less common in California herds (33%) [48]. Such practice contributes to an increase in pathogenic bacteria in the uterus [28]. It also increases the risk of damage to the endometrium (uterine hemorrhages, hematomas and vascular thrombi) and suppresses uterine leukocyte phagocytosis [49]. Moreover, it is difficult to ensure that the entire placenta has been removed, with necrotic portions left behind, further contributing to bacterial invasion of the now-damaged endometrium [7]. Such practice needs to be discouraged except in the case where the manual extraction of the placenta does not require the manipulation of caroncules and cotyledons.

A general examination is always carried out by only 38% of the respondents. However, 47.5% always take the temperature, a parameter generally recognized as a criterion for the choice of an appropriate treatment approach [1,50,51]. Involving the farmer in taking the temperature would help to apply it more systematically. In this context, the relational analysis revealed that other European farmers took temperature more frequently than French and Wallonian farmers.

Cow-side tests to detect ketosis are very often used by a few veterinarians. However, such tools can be very useful in deciding a treatment with propylene glycol or glycerol [52].

According to our results, whether the cow has or does not have a fever seems important to the decision of whether intrauterine or/and systemic approaches should be used to treat RFM. Intrauterine treatment is more often used in the case of “no fever”, and a systemic treatment is more often used in the case of “fever”. Such differences have also been observed in another recent survey (47.6% vs. 10.7% in dairy cattle and 24.2% vs. 24.2% in beef cattle [41]). The simultaneous use of both routes was largely more often used in the case of fever (70.4%) than in the absence of fever (10.2%). Such observations have also been observed in dairy (61.8% vs. 6.1%) and beef cattle (56.3% vs. 29.5% [41]). In another study, 93.3% of veterinarians treated cows with fever via the parenteral route [46]. Fever is only an indicator of postpartum inflammation. Additional clinical signs, such as uterine discharge, are necessary to identify uterine bacterial infection [53]. In one study, systemic antibiotics alone were as effective as systemic antibiotics combined with intrauterine treatment [54]. In Belgium, 7.4% of respondents treating beef cows with RFM with fever use the intraperitoneal route. This route is not registered, and its effectiveness and pharmacokinetics are unknown. Therefore, this injection route seems hard to justify [41]. According to different studies, it seems that local antibiotics, typically given as uterine infusions or boluses, have not been shown to reduce the incidence of metritis or improve fertility [2,21,26,55,56]. Moreover, such a local approach could interfere with the necrotizing process that is responsible for the eventual release of RFM [29]. Moreover, systemic antibiotic treatment of RFM in the absence of systemic illness has not been proven to have any beneficial effects compared to a selective treatment [26]. As concluded by [41], a considerable proportion of cases where this antibiotic is used and the associated cost of the treatment can be avoided without deleterious effects on the animal’s well-being and performance by better selecting the cows that truly need antibiotic therapy. Constable [57] already proposed an approach to limit antimicrobial therapy to cows with clinical signs of puerperal metritis.

Most respondents in our study use, sometimes or never, intrauterine treatment with antiseptic, iodine or essential oil solutions. The use of povidone–iodine (2%) improves [58] or not [59,60] fertility in the case of endometritis. The positive effect of a solution of carvacrol, a phenol produced by aromatic plants, compared to a solution of povidone–iodine, has been shown for the treatment of puerperal metritis [61].

Some alternative intrauterine treatments, like collagenase injection into the umbilical cord [62] or ozone [38,40,63], have been proposed. Ozone has germicidal and fungicidal properties [37,64] and stimulates immunity through the induced production of cytokines [39]. Ozone foam can be recommended as a non-antibiotic intrauterine treatment for cattle with RFM [38,40]. In our survey, those treatments are more often used in Sweden, but no statistical difference was observed because of the low frequency of use.

In order of preference, tetracyclins, cefapirin and penicillins are the three main antibiotics used in utero, and penicillin is the main antibiotic used for systemic treatments. In Switzerland, tetracycline boluses are used in > 80% of RFM cases [46]. In Belgium, the most frequently used antibiotic molecules to systemically treat RFM were β-lactams (benzylpenicillin, amoxicillin and ampicillin), followed by tetracyclines and trimethoprim-sulfadiazine [41]. Cephalosporins are not used frequently. Here, in a recent survey carried out in Belgium [41], third-generation cephalosporins were used by 2.8% and fourth-generation cephalosporins were used by 1.4%, respectively. In California dairies, ceftiofur products are used for cows with RFM in 66% of dairies [48]. An USDA-NAHMS [65] report describes cephalosporin as the primary antimicrobial class used for reproductive disease in dairy farms in the United States, followed by penicillin and tetracycline. In the Midwest and Northeast of the USA, we observed a similar frequency of use for ceftiofur products (41.4%) or penicillin (43.4%) to treat metritis and RFM [66].

There is a great variety of antibiotic treatment protocols for RFM. Some are used as intrauterine antimicrobial agents (oxytetracycline, ampicillin and cloxacillin) and others are used as systemic antibiotics (penicillin, amoxicillin, ampicillin, oxytetracycline and ceftiofur). Their effect and efficacy are still debated [22]. Opposite observations have been noted after various combinations of intrauterine infusions of tetracyclines for the treatment of RFMs [56]. The combination of amoxicillin via the parenteral route plus an intrauterine infusion of oxytetracycline, was more efficacious than using only the parenteral route. The efficacy of tetracyclines could depend on the duration of administration, dosage, regimen or anti-inflammatory effects of tetracyclines [67].

PGF2α and oxytocin are sometimes used (48%) or always used (14.5%) by 34.0% and 3.4% of the respondents, respectively. In a previous survey [41],63.5 to 70.3% of the respondents declared they never or rarely used natural or synthetic PGF2α and 55,4 to 76,2% oxytocin or long-acting oxytocin. In Switzerland, between 37% and 40% of the practitioners use oxytocin or PGF2α [46]. In the USA, very few dairies (5 out of 45) use oxytocin in cases of RFM [48]. According to several studies, it seems that oxytocin or PGF2α are not effective for the prevention or treatment of RFM [21,34,35,36,68,69]. Conversely, only one work observed a reduction (10.9 vs. 24.6%) in the risk of placental retention after the injection of 30 IU of oxytocin [70]. Oxytocin (50 IU) and carbetocin (0.35 mg) seem to have the same uterotonic effects in healthy cows [71]. PGF2α and oxytocin play a role in uterine contraction and could be effective in treating RFM in cases of uterine atony. However, such a situation accounts for a very small percentage of retained placenta cases [10]. Finally, some Chinese studies proposed the use of traditional herbs to reduce the incidence of RFM [72,73].

In a recent study, the prevalence of retained placenta was negatively associated with the serum calcium concentration [74]. Moreover, hypocalcemia reduces dry matter intake (DMI) and compromises energy metabolism [75]. It is also an important risk factor for metritis [76] due to its negative effect on the activity of immune cells [77]. This justifies the occasional use of Ca therapy by half of veterinarians in our study and by 53 to 57% in another survey conducted in Switzerland [46]. However, this strategy needs to be evaluated further. Also, hypocalcemia is a separate postpartum pathology that can coincide with RFM.

Our results revealed that half of the practitioners sometimes used a nonsteroidal anti-inflammatory drug (NSAID). In a study conducted in Switzerland, 79,1% of the practitioners declared using NSAIDs in the presence of fever, and 31.9% declared their use in the absence of fever [46], respectively. According to a recent review, the potential benefits of such a treatment depend on the type of drug employed and its dosage and administration mode [78]. Salicylates and flunixin meglumine show significant side effects like increased risk of retained placenta, culling or metritis. Conversely, a positive effect on milk production seems to be plausible. Nevertheless, it seems that this routine use should be carefully evaluated and that it could possibly be associated with early diagnosis of high inflammation status.

## 5. Conclusions

Our study, which was conducted across several European countries, confirms the necessity of improving and rationalizing the diagnostic and therapeutic approaches related to RFM in the context of different measures to reduce the important problem of antibiotic resistance.

## Figures and Tables

**Table 1 animals-14-01042-t001:** The table shows the prevalence and etiology of RFM (the number of answers in each class is shown in parentheses).

RFM definition (h) (682)	<6	12	24	>24		
	5.4	38.9	48.4	7.3		
Speculation concerned (556)	No difference	Dairy	Beef			
	76.1	10.3	13.6			
Estimated prevalence	<5%	5–10%	11–15%	>15%	I don’t know	
Dairy (609)	22.0	46.6	17.1	4.3	10.0	
Beef (563)	49.9	26.8	5.2	0.9	17.2	
Suspected etiology	Male calf	C-section	BCS < 3	Dead calf	Dystocia	BCS > 4
Dairy	3.8 (184)	9.1 (241)	15.3 (249)	25.8 (314)	26.8 (351)	30.0 (350)
Beef	3.8 (185)	14.4 (284)	9.7 (236)	24.4 (324)	26.4 (349)	18.0 (272)
	Induction	Lack	Excess	Twinning	Abortion	
Dairy	34.0 (306)	35.0 (363)	36.2 (348)	48.6 (562)	59.0 (551)	
Beef	36.0 (297)	36.0 (367)	25.3 (288)	47.1 (501)	59.8 (527)	

% of veterinarians who chose the factor as the primary cause. Excess: excess of minerals before calving; Lack: lack of trace elements before calving; BCS: body condition score.

**Table 2 animals-14-01042-t002:** The table shows the methods used for RFM diagnosis (the number of answers in each class is shown in parentheses).

	Never	Sometimes	Always
Milk (cetone bodies or NEFA)	86.4	12.4	1.2
Urine(cetone bodies or NEFA)	80.3	16.9	2.8
T° by the farmer	60.9	35.9	3.1
Blood (cetone bodies or NEFA)	75.9	20.2	3.9
Dry (vaginal exploration without washing the vulva)	57.7	21.1	21.2
Manual palpation of the uterus	46.6	29.7	23.7
Water use for washing before vaginal exploration	52.1	22.8	25.1
Antiseptic use for washing before vaginal exploration	45.1	24.3	30.6
General examination	12.5	49.5	38.0
T° by the veterinarian	16.8	35.7	47.5
Vaginal exploration	15.7	13.2	71.1

**Table 3 animals-14-01042-t003:** The table shows the treatments (the number of answers in each class is shown in parentheses).

Treatments by Intrauterine Route	Never	Sometimes	Always
Tetracyclins (640)	22.5	33.4	44.1
Cefapirin (595)	25.7	33.5	40.8
Penicillins (576)	56.6	29.0	14.4
Others (510)	91.5	6.1	2.4
Rifaximin (529)	91.5	6.2	2.3
Antiseptic solutions (622)	74.7	24.3	1.0
Essential oils (489)	88.3	10.8	0.9
Iodine solution (<200 mL) (615)	85.5	13.7	0.8
Aminosides (503)	96.2	3.6	0.2
Kanamycin (518)	99.4	0.6	0.0
Ozone (389)	99.0	1.0	0.0
Collagenase in umbilical artery (382)	99.5	0.5	0.0
Antibiotics used in a systemic way	Never	Sometimes	Always
Penicillins (647	11.3	66.6	22.1
Tetracyclins (639)	43.2	50.7	6.1
Cephalosporines G3, G4 (635)	53.1	43.4	3.5
Sulfamids (635)	80.8	17.2	2.0
Others (631)	88.4	10.0	1.6
Cephalosporines Generation 1 and 2 (631)	84.4	15.1	0.5
Macrolids (633)	90.4	9.3	0.3
Quinolones (636)	84.9	15.1	0.0
Non-antibiotic systemic treatments	Never	Sometimes	Always
PGF2a (642)	37.5	48.0	14.5
NSAID (575)	41.6	46.1	12.3
Homeopathy (617)	64.5	29.7	5.8
Oxytocin (635)	62.4	34	3.6
Carbetocin (495)	90.5	7.7	1.8
Others (560)	90.2	8.9	0.9
Ca perfusion (632)	59.3	40.1	0.6
Corticoïds (630)	75.7	23.8	0.5
Acupuncture (597)	97.7	2.3	0.0

**Table 4 animals-14-01042-t004:** The table shows the ordinal regression analysis.

Parameter	Variable		OR	95% CI		*p*
The farmer has already taken the temperature of the cow (Pseudo R^2^: 0.42)	Country	France	Referent			
		Wallonia	1.70	0.95	3.04	0.076
		Flanders	13.08	6.61	25.89	<0.001
		Austria	22.00	9.79	49.40	<0.001
		Sweden	16.86	10.41	27.33	<0.001
		Netherlands	56.54	25.15	127.10	<0.001
No treatment if the cow has no fever (Pseudo R^2^: 0.14)	Country	France	Referent			
		Wallonia	0.52	0.34	0.82	0.005
		Flanders	0.76	0.42	1.37	0.362
		Austria	0.37	0.18	0.77	0.008
		Sweden	0.56	0.38	0.82	0.003
		Netherlands	0.04	0.01	0.13	<0.001
	Experience	>10 years	Referent			
		10–20	0.54	0.37	0.77	0.001
		<10 years	0.56	0.39	0.80	0.002
Intra-uterine treatment with cefapirin (Pseudo R^2^: 0.65)	Country	France	Referent			
		Wallonia	0.62	0.38	1.03	0.066
		Flanders	0.0079	0.0027	0.0226	<0.001
		Austria	0.0017	0.0005	0.0065	<0.001
		Sweden	0.0021	0.0008	0.0058	<0.001
		Netherlands	0.0005	0.0001	0.004	<0.001
	Bovine practice	<20%	Referent			
		20–80%	0.47	0.32	0.69	<0.001
		>80%	0.35	0.17	0.72	0.004
Ocytocine (Pseudo R^2^: 0.19)	Country	France	Referent			
		Wallonie	2.44	1.47	4.06	0.001
		Flanders	4.07	2.14	7.74	<0.001
		Austria	14.08	6.61	29.96	<0.001
		Sweden	5.29	3.42	8.19	<0.001
		Netherlands	6.63	3.42	12.81	<0.001
Calcium perfusion (Pseudo R^2^: 0.19)	Country	France	Referent			
		Wallonia	0.57	0.32	1.02	0.06
		Flanders	1.95	1.03	3.68	0.04
		Austria	11.21	4.69	26.82	<0.001
		Sweden	4.74	3.08	7.32	<0.001
		Netherlands	3.26	1.68	6.33	<0.001
Non-Steroidal Anti-Inflammatory Drugs (Pseudo R^2^: 0.41)	Country	France	Referent			
		Wallonia	1.43	0.85	2.39	0.176
		Flanders	6.29	3.22	12.28	<0.001
		Austria	32.62	14.63	72.75	<0.001
		Sweden	7.85	4.92	12.55	<0.001
		Netherlands	46.29	21.56	99.38	<0.001
	Experience	>10 years	Referent			
		10–20	2.42	1.58	3.70	<0.001
		<10 years	4.73	3.10	7.21	<0.001

## Data Availability

The data presented in this study are available on request from the corresponding author. The data are not publicly available due to their large amount.

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
