# Peer review of "Propaedeutic and Therapeutic Practices Used for Retained Fetal Membranes by Rural European Veterinary Practitioners"

_animals, 2024, doi:10.3390/ani14071042_

Round 1

Reviewer 1 Report

Comments and Suggestions for Authors

Propaedeutic and therapeutic practices..

The study covers an important subject and is worth being published. There are things to add / correct, though.

Please change throughout the whole manuscript:

-          Veterinarian instead of vet

-          Examination instead of exam

-          Please carefully check how to use the word  respectively: it can be used at the end of a sentence after the reader knows the options – please correct throughout manuscript  

-          Please replace throughout manuscript PGF2a with PGF2a

-          please indicate always, sometimes never throughout text in quotations marks or italics

Title: could be changed to Diagnostic and therapeutic practices used for retained fetal membranes by European rural practitioners

Data analysis: as the study is probably not representative, the reviewer recommends only descriptive statistics

Mat&Meth: please define percentage always, sometimes, never

Mat & Meth: please indicate threshold for fever

Mat & Meth: please describe why these countries were selected

Page 1, line 18: replace propaedeutic with diagnostic

Page 1, line 21: interspace before our; confirms

Page 1, line 23 and 35: replace antibioresistance with antibiotic resistance

Page 1, line 29: has

Page 1, line 31: first time write out the word: intrauterine (IU)

Page 1, line 32: oxytocin

Keywords: replace dairy cattle, beef cattle by cattle and propaedeutic by diagnostic

Page 2, line 45: delete rate; reproductive performances

Page 2, line 49: …on 50 citations .. but you indicate only two ?? explain

Page 2, line 70: [21] Dyrendahl is from 1977 – I’d delete this reference

Page 3, line 117: % in countries = 100.1%, please correct

Page 3, line 124: please change sign, I suppose its < 6h? Then indicate ranges 6-12h etc so that categories are distinct

Page 3, line 126: Practitioners indicate that

Page 3, line 135: write out C-section

Page 4: RFM definition Table 1: compare to text and see comment for page 3, line 124

Page 4, line 143: antiseptic solution (AS) (30.6%) or…

Page 4, line 155: sentence unclear, is it meant: very few veterinarians treat the animals if the animals do not present fever

Page 5, line 162: …the most often used..

Table 3: collagenase used intrauterine or directly to umbilical artery? Which kind of uterotonic oral solution?

Page 7, line 217: does not

Page 7, line 231. Ref [37] 93.3% of cows with fever were treated with parenteral antibiotics in >80% of the cases

Page 8, line 275 and further: please delete the whole passage on puerperal metritis

Page 9, line 300: replace prevent with reduce the incidence

Page 9, lines 301-307: please mention, that Ca-infusion is very often not a treatment of RFM, but a frequent illness post partum in dairy cattle, coincident, mostly not causal

Page 9, line 312: reference missing

Comments on the Quality of English Language

Do not use abbreviations (exam, vet) and write words out the first time you use them (intrauterine, C-section); check the use and correct place of the word "respectively" within a sentence.

Author Response

Reviewer1:

Dear Sir/madame,

We are very grateful for the time you gave to carefully consider our manuscript. All the comments you made were taken into consideration. Please, find bellow our answers.

Comment 1

Page 1, line 18: replace propaedeutic with diagnostic

Answer

The word was replaced. See line 18 of the revised manuscript.

Comment 2

Page 1, line 21: interspace before our; confirms

Answer

Interspace was added. See page 1 line 21 of the revised manuscript.

Comment 3

Page 1, line 23 and 35: replace antibioresistance with antibiotic resistance

Answer

It was replaced. See line 23 of the revised manuscript.

Comment 4

Page 1, line 29: has

Answer

It was replaced. See line 29 of the revised manuscript.

Comment 5

Page 1, line 31: first time write out the word: intrauterine (IU)

Answer

It was replaced. See line 31 of the revised manuscript.

Comment 6

Page 1, line 31: first time write out the word: intrauterine (IU)

Answer

It was replaced. See line 31 of the revised manuscript.

Comment 7

Page 1, line 32: oxytocin

Answer

It was replaced. See line 32 of the revised manuscript.

Comment 8

Keywords: replace dairy cattle, beef cattle by cattle and propaedeutic by diagnostic

Answer

Answer: It was replaced. See keywords of the revised manuscript.

Comment 9

Page 2, line 45: delete rate; reproductive performances

Answer

It was deleted. See page 2 line 45 of the revised manuscript.

Comment 10

Page 2, line 49: …on 50 citations .. but you indicate only two ?? explain

Answer

The frequency of RFM has been reviewed by Kelton et al. in 1998, we have add the reference.

Comment 11

Page 2, line 70: [21] Dyrendahl is from 1977 – I’d delete this reference

Answer

Attending the neagtaive effect of such practice, we have preferred to suppress this reference.

Comment 12

Page 3, line 117: % in countries = 100.1%, please correct

Answer

It was verified that Belgium has 20,6% of respondents and not 20,7. It is now 100%. See page 3, line 117

Comment 13

Page 3, line 124: please change sign, I suppose its < 6h? Then indicate ranges 6-12h etc so that categories are distinct

Answer

It was corrected. See page 3, line 123-124 of the revised manuscript.

Comment 14

Page 3, line 126: Practitioners indicate that

Answer

It was corrected. See page 3, line 125 of the revised manuscript.

Comment 15

Page 3, line 135: write out C-section

Answer

It was written out as recommended. See line 134 of the revised manuscript

Comment 16

Page 4: RFM definition Table 1: compare to text and see Comment for page 3, line 124

Answer

Text was corrected according to the table. See lines 123-124 of the revised manuscript

Comment 17

Page 4, line 143: antiseptic solution (AS) (30.6%) or…

Answer

“Or” was deleted. See line 142 of the revised manuscript.

Comment 18

Page 4, line 155: sentence unclear, is it meant: very few veterinarians treat the animals if the animals do not present fever

Answer

The expression was replaced by: “Most veterinarians treat cows with RFM even if animals do not present fever”. See line 154 of the revised manuscript.

Comment 19

Page 5, line 162: …the most often used..

Answer

It was added. See line

Comment 20

Page 5, line 162: …the most often used..

Answer

It was added. See line 161 of the revised manuscript

Comment 21

Table 3: collagenase used intrauterine or directly to umbilical artery?

Which kind of uterotonic oral solution?

Answer

Collagenase use in the umbilical artery has been added

This proposal (uterotonic solution) has been deleted

Comment 22

Page 7, line 217: does not

Answer

It was replaced. See line 216 of the revised manuscript

Comment 23

Page 7, line 231. Ref [37] 93.3% of cows with fever were treated with parenteral antibiotics in >80% of the cases

Answer

The information with the reference ware added. Please see lines 233-234 of the revised manuscript

Comment 24

Page 8, line 275 and further: please delete the whole passage on puerperal metritis

Answer

The passage about puerperal metritis was deleted. See lines 275-283 of the revised manuscript

Comment 25

Page 9, line 300: replace prevent with reduce the incidence

Answer

It was replaced. See line 296 of the revised manuscript

Comment 26

Page 9, lines 301-307: please mention, that Ca-infusion is very often not a treatment of RFM, but a frequent illness post partum in dairy cattle, coincident, mostly not causal

Answer

It was mentioned: “Also, hypocalcemia is a separate post-partum pathology that can coincide with RFM”. Please see lines 304-305 of the revised manuscript.

Comment 27

Page 9, line 312: reference missing

Answer

The reference was moved to the end of the sentence. Please see line 310 of the revised manuscript

Other Comments:

Comment

Do not use abbreviations (exam, vet)

Answer

Abbreviations were replaced by the whole words. Please see the manuscript

Comment

Write words out the first time you use them (intrauterine, C-section)

Answer

Words were written out the first time.

Comment

Check the use and correct place of the word "respectively" within a sentence

Answer

Done

Reviewer 2 Report

Comments and Suggestions for Authors

Well presented and expounded article, good research, good conclusion.

The scientific work is well expressed, well written, and well articulated. The research topic covers very frequent issues in the field clinic and addresses the major methods used and explained in both textbooks and bibliographical references. Very interesting is the statistical analysis and results regarding the ages of the veterinarians examined, this factor could be a cue for future research, could put attention on different practices over the years. the bibliographical references are comprehensive,the conclusions reflect the results obtained and are supported adequately.

The work represents areas of interest and study for buiatrists in the field.

Author Response

Reviewer 2:

Dear Sir/madame,

We are very grateful for the time you give to carefully consider our manuscript.

Reviewer 3 Report

Comments and Suggestions for Authors

Authors made a survey to describe how rural bovine practitioners act when diagnosing and treating a fetal membranes retention. 

In my humble opinion, this study presented a non-validated survey, which difficults its impact in science. Moreover, they did not apply normality test, nor correlation ones. 

Moreover, it has been completely unrespectful to present data in the way they did: using "," in stead of "." in numbers, different formats in tables, bibliography has a lot of mistakes and it is wrongly referenced throughout the manuscript, they use acronyms without first explanation, etc. 

Publishing scientific results must be rigorous and correct. Besides, experimental design must be appropriate. 

I think that results could be interesting but authors must work to improve the manuscript. However, I wonder if this data are in line with Animals scope.

Comments on the Quality of English Language

Moderate editing of English language required

Author Response

Reviewer 3:

Dear Sir/madame,

We are very grateful for the time you gave to carefully consider our manuscript. All the comments you made were taken into consideration. Please, find bellow our answers.

Comment

This study presented a non-validated survey:

Answer

In fact the survey lacks the validation, because it was done in many languages to cover the different countries and wanted to survey the maximum number of veterinarians in as many countries as possible.

Comment

normality test, nor correlation ones

Answer

Concerning the statistical analysis. We just described the practices as it is the main aspect of this review. Moreover, we used ordinal logistic regression to study the influence of different demographic factors on veterinarians’ practices diagnosing and treating this pathology.

Comment

 Using "," in stead of "." in numbers,

Answer

All "," were replaced by "." in numbers

Comment

English

Answer

Improvements were applied to language

Round 2

Reviewer 3 Report

Comments and Suggestions for Authors

Authors did not accomplish my suggestions, they just had a short answer for each. Nor even the ones of format or bibliography...

I still thinking that manuscript should be improved substantially prior to publication.

Comments on the Quality of English Language

-